# How accurate are WorldPop-Global-Unconstrained gridded population data at the cell-level?: A simulation analysis in urban Namibia

Dana R. Thomson [1,2]*, Douglas R. Leasure [3¤a], Tomas Bird [3¤b], Nikos Tzavidis [2], Andrew J. Tatem [3]

**1** Faculty of Geo-Information Science & Earth Observation, University of Twente, Enschede, The Netherlands, **2** Department of Social Statistics and Demography, University of Southampton, Southampton, United Kingdom, **3** WorldPop, Geography and Environmental Science, University of Southampton, Southampton, United Kingdom

¤a Current address: University of Oxford, Leverhulme Centre for Demographic Science, Oxford, United Kingdom
¤b Current address: NorthWest Atlantic Fisheries Centre, Department of Fisheries and Oceans, St. John's, Canada

* d.r.thomson@utwente.nl

**Data Availability Statement:** All relevant data are within the paper and its Supporting Information files.

## Abstract

Disaggregated population counts are needed to calculate health, economic, and development indicators in Low- and Middle-Income Countries (LMICs), especially in settings of rapid urbanisation. Censuses are often outdated and inaccurate in LMIC settings, and rarely disaggregated at fine geographic scale. Modelled gridded population datasets derived from census data have become widely used by development researchers and practitioners; however, accuracy in these datasets are evaluated at the spatial scale of model input data which is generally courser than the neighbourhood or cell-level scale of many applications. We simulate a realistic synthetic 2016 population in Khomas, Namibia, a majority urban region, and introduce several realistic levels of outdatedness (over 15 years) and inaccuracy in slum, non-slum, and rural areas. We aggregate the synthetic populations by census and administrative boundaries (to mimic census data), resulting in 32 gridded population datasets that are typical of LMIC settings using the WorldPop-Global-Unconstrained gridded population approach. We evaluate the cell-level accuracy of these gridded population datasets using the original synthetic population as a reference. In our simulation, we found large cell-level errors, particularly in slum cells. These were driven by the averaging of population densities in large areal units before model training. Age, accuracy, and aggregation of the input data also played a role in these errors. We suggest incorporating finer-scale training data into gridded population models generally, and WorldPop-Global-Unconstrained in particular (e.g., from routine household surveys or slum community population counts), and use of new building footprint datasets as a covariate to improve cell-level accuracy (as done in some new WorldPop-Global-Constrained datasets). It is important to measure accuracy of gridded population datasets at spatial scales more consistent with how the data are being

**Funding:** Dana R. Thomson was funded by the Economic and Social Research Council (ESRC) grant number ES/5500161/1 (more information at https://esrc.ukri.org/). ESRC played no role in the design, analysis, decision to publish, or preparation of this manuscript.

**Competing interests:** The authors have declared that no competing interests exist.

applied, especially if they are to be used for monitoring key development indicators at neighbourhood scales within cities.

## Introduction

Small area population counts, especially in low- and middle-income countries (LMICs), provide essential denominators for health, economic, and development indicators [1]. For example, small area population counts are used to calculate vaccination coverage rates [2], understand health service utilisation [3], and estimate infection rates of malaria, COVID-19, and many other health conditions [4]. Spatially-detailed and time-sensitive population counts are also essential to monitor and understand the accelerated pace of urbanisation in LMICs compared to HICs. Ninety percent of global population growth in the next 30 years is expected to occur in African and Asia cities alone [5], which means it is vital to monitor population trends across diverse LMIC cities with respect to economic development, human impacts on biodiversity and environment, and the changing climate [6,7]. Authoritative population data are traditionally collected via a national census. Censuses are generally collected every ten years, though one in ten LMICs has not held a census in the last 15 years [8], and some national censuses have poor data quality due to negligence (e.g., [9,10]) or deliberate miscounting of sub-populations for political purposes (e.g., [11–13]). Due to increasing rates of mobility and urbanisation worldwide, the urban poorest–especially in LMIC cities–are increasingly difficult to count as more people take-up residence in informal settlements or atypical housing locations (e.g., shops) [14].

In the absence of updated, fine-scale census data, many policy-makers, urban planners, researchers, and service providers have turned to gridded population estimates as a source of population counts in their work. Gridded population data are viewed by data producers and users as meeting a global development challenge to "leave no one off the map" and thus leave no one behind [15]. However, performing accuracy assessments of gridded population datasets at the scale at which they are applied (e.g., neighbourhood, grid cell) poses a conundrum; reliable fine-scale population counts are generally not available where they are needed most [16], and users often turn to gridded population estimates when census counts are excessively outdated or untrustworthy [14]. Despite these challenges, it is imperative to understand if, and how, census inaccuracies propagate through gridded population datasets, especially with respect to vulnerable populations.

Briefly, gridded population data provide estimates of the total population in small grid cells, and are derived with geo-statistical methods using population counts and spatial datasets [16]. "Top-down" gridded population estimates have been available for roughly 15 years and disaggregate census or other complete population counts from areal units (e.g., 3rd-, 4th-, or 5th-level administrative units) to grid cells (e.g., 30x30m, 100x100m, 1x1km) [14]. The simplest models assume a uniform distribution of population within areal units (i.e., GPW [17,18], GHS-POP [19,20], HRSL [21]), while the most complex models use spatial covariates to inform spatial disaggregation from the areal unit to grid cells (i.e., WorldPop [22,23], LandScan [24,25], WPE [26]). To estimate gridded population figures beyond the year of the last census; birth, migration, and death rates are used to project new population totals by areal unit [27]. "Bottom-up" gridded population estimates are derived from micro-census population counts in a sample of areas, or from assumptions about the average household size, and have only recently been developed [28,29]. Read papers by Leyk and colleagues (2019) and Thomson and colleagues (2020) for detailed descriptions and comparisons of gridded population datasets [14,16].

The accuracy of "top-down" gridded population data is generally calculated at the scale of the input population areal units because these are the finest-scale population counts available to the data producers. A number of factors contribute to gridded population model accuracy including: (1) the modelling algorithm itself, (2) inaccuracy of the input population data, (3) the geographic scale of the input population data (e.g., census tracts versus districts), (4) the age, accuracy, completeness, and type of ancillary data, (5) the nature of the relationship between ancillary data and population density, and (6) the geographic scale of the output grid. Of these, the two strongest predictors of accuracy (at the scale of areal units) in top-down gridded population models are the resolution and age of the input population data [30]. Among top-down gridded population datasets, the WorldPop-Global-Unconstrained Random Forest model was among the best documented and most accurate gridded population models available at the time of this analysis in 2017–2019 [22,31]. Specifically, the model code [32] and pre-processed model covariates [33,34] were publicly available enabling reproducibility and evaluation. WorldPop-Global-Unconstrained and its preceding data products (AfriPop, Asia-Pop, and AmeriPop) result in estimates for all land areas; however, a new WorldPop-Global-Constrained dataset was published in 2020 limiting population estimates to cells with buildings or built-up features [35].

To evaluate cell-level accuracy of gridded population data, actual population counts are needed for each grid cell or in finer-scale units such as household point locations. Few censuses in LMICs collect household latitude-longitude coordinates, and where these censuses exist, the data are extremely sensitive and difficult to obtain. Furthermore, even the best census data might be problematic because vulnerable sub-populations including homeless and nomadic populations are supposed to be counted separately in special enumerations. Unfortunately, though, under-resourced statistical offices are often not able to perform these counts [36], and some censuses do not include certain refugee or internally displaced populations [37]. To ensure that this analysis of cell-level accuracy did not exclude the urban poorest and other hidden populations, we chose to simulate a realistic population in a LMIC setting. It was important that the synthetic population was located in a real-world location so that actual covariate datasets–with their own imperfections–could be used to generate realistic gridded population datasets. We adapted methods outlined by Thomson and colleagues (2018) for simulating a geo-located realistic household population, and added classification of urban households by slum/non-slum area in a final step to focus this analysis on dynamic, complex LMICs cities where inaccuracies in gridded data are likely to propagate [38].

This paper describes how we evaluated the cell-level accuracy of 32 simulated 100x100m WorldPop-Global-Unconstrained gridded population datasets which reflect realistic levels of census (1) outdatedness (0-, 5-, 10-, and 15-years outdated), (2) inaccuracy (none, low, middle, and high missing population counts), and (3) two administrative-level aggregations of the population in an urban LMIC setting. This is among the first assessments of cell-level accuracy of a gridded population dataset in a LMIC setting. While the methods and approach outlined here to evaluate cell-level accuracy (developing a realistic synthetic population, and from this, deriving several realistic versions of census data) were applied to just one gridded population dataset, they could be applied to other gridded population data products used for development monitoring and decision-making.

## Methods

### Setting

We chose to simulate a population in Khomas, Namibia–in which the vast majority of residents reside in Windhoek, the capital–because the government has produced numerous high-

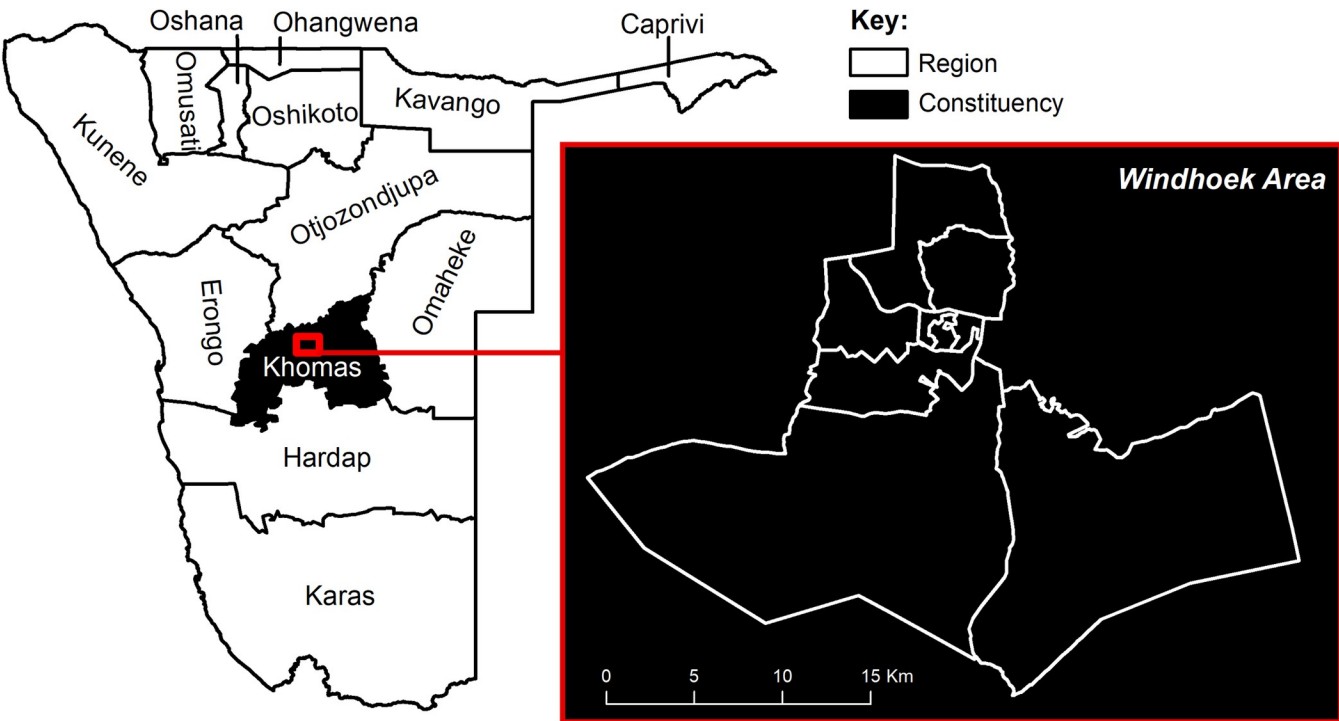

**Fig 1. Location of Khomas region in Namibia, and of constituencies in Windhoek area.** Source: Constituency boundaries publically available from https://gadm.org/.

quality population datasets [39], and Windhoek's population is incredibly dynamic (Fig 1). Namibia, like some other countries that inherited colonial boundaries, placed restrictions on freedom of movement until independence in 1990 [40]. After independence, vast numbers of people migrated to Windhoek, exaggerating rural-to-urban migration patterns observed globally during this time period [41,42]. Windhoek is also a destination for immigrants from neighbouring countries including financially unstable Zimbabwe [42,43]. The population of the Windhoek metropolitan area grew by a staggering 37% between the 2001 and 2011 censuses [39], with much of that growth in informal settlements [40].

## Simulation overview

To simulate realistic gridded population datasets for Khomas, Namibia, we (a) simulated a "true" synthetic 2016 population geo-located to realistic manually-generated household point locations; (b) introduced realistic outdatedness by removing households in 2011, 2006, and 2001; (c) introduced realistic inaccuracies among urban-slum, urban-non-slum, and rural sub-populations; and (d) aggregated these 16 simulated population scenarios into two geographic areal units (census EA and constituency) to generate 32 realistic census datasets. These 32 realistic census datasets were consequently used to model 32 realistic WorldPop-Global-Unconstrained 100x100m gridded population datasets. This workflow is summarised in Fig 2 and detailed below.

## Simulating a "true" synthetic 2016 population geo-located to household latitude-longitude points

To simulate a realistic population in Khomas, Namibia, we used all of the same population inputs and spatial auxiliary datasets as Thomson and colleagues (2018) [38]. Broadly, this

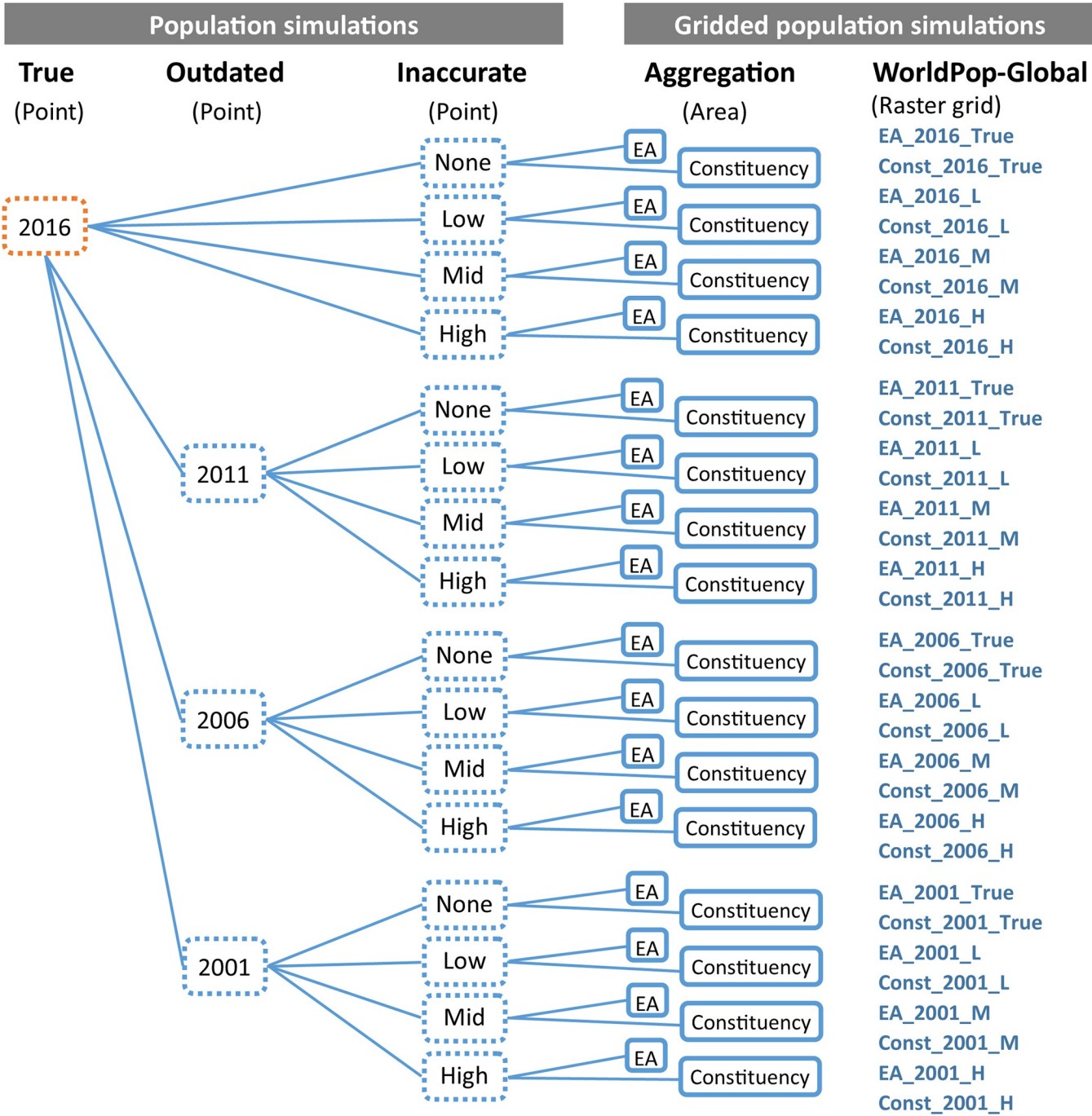

**Fig 2. Summary of the population and gridded population simulation workflow.** (1) Simulate a realistic population geo-located to realistic building point locations, (2) simulate three periods of outdatedness by removing households at point locations not present on satellite imagery in earlier years, (3) simulate low/middle/high census inaccuracy by removing points at random from rural, urban-slum, and urban-non-slum household types, (4) aggregate to 922 census enumeration areas (EAs) and 10 constituencies (admin-2), (5) generate 100x100m gridded population datasets in raster grid format using WorldPop-Global-Unconstrained approach and WorldPop-Global spatial covariates.

involved the creation of three datasets—modelled surfaces of household types, manually digitised building point locations, and synthetic (simulated) households—then linked synthetic households to point locations based on the household type probability surfaces.

1. Modelled surfaces of household types. Household types were defined from Namibia 2013 Demographic and Health Survey (DHS) data using k-means analysis with variables that were also present in the Namibia 2011 census (e.g., improved sanitation facilities, gender of head of household). Next, probability surfaces of these household types were created using a Random Forrest model and spatial covariates to interpolate the likelihood of a given household type across Namibia between DHS survey locations [38]. The probability surfaces of "urban poor" and "urban non-poor" household types were manually adjusted due to high misclassification. These adjustments were made by manually assigning the proportion of households in each census enumeration area (EA) that appeared to be located in areas of small disorganised buildings based on visual inspection of 30m Quickbird satellite imagery.

2. Synthetic households. Separately, we modelled a synthetic population of individuals nested within households across Khomas from Namibia 2011 census microdata using an iterative proportional fitting model and conditional annealing [44].

3. Building locations. A third set of data, building point locations, were manually digitised from 2014–2016 30cm Quickbird imagery in ArcGIS 10.

To link synthetic households with building locations, we calculated the most likely household type of each synthetic household using k-means analysis scores. Next, we iteratively assigned synthetic households (2) to building point locations (3) based on the probability of each household type at a given building point (1). Finally, using the manually classified EAs (with our estimated portion of urban poor households), we classified all urban households as being located in either a slum or non-slum area. All of these steps are detailed in Supplement 1 and the paper by Thomson and colleagues (2018) [38]. This simulated population is meant to represent a realistic "true" synthetic reference population for 2016.

**Simulating realistic outdatedness of Khomas census population.** To simulate population outdatedness in Khomas, we imported the above 2016 synthetic population household point locations into Google Earth, and used the software's historical Maxar and SPOT imagery (40cm) to flag all buildings that were not present in 2011, 2006, and 2001 imagery. The oldest imagery available at 40cm resolution in Google Earth was from 2004, so we used some judgement to flag buildings that looked recently built in 2004 (e.g., bare fresh soil) and assumed they were not present in 2001. During this exercise, we ensured that the number of household coordinates in each constituency matched the number of households reported in the 2001 and 2011 Population and Housing Census final reports to ensure that both patterns and degree of outdatedness were realistic [39] (Fig 3). The synthetic population is provided in Supplement 2 and is comparable to the Oshikoto, Namibia 2016 synthetic population created by Thomson and colleagues [38].

**Simulating realistic levels of under-count inaccuracy in censuses.** To identify realistic levels of under-counts among urban-slum, urban-non-slum, and rural populations in LMIC censuses, we reviewed the scientific and grey literature. The review included census post enumeration surveys (PESs) in 108 LMICs listed by the UN Statistical Division Census Programme website [8], and a systematic search in PubMed and Scopus of articles published between January 1, 1990 and February 28, 2017 using the following search criteria: "census AND (listing OR enumerat* OR count OR coverage OR miss*) AND (nomad* OR pastoral* OR refugee OR displaced OR migrant OR slum OR poorest OR unregistered OR homeless OR [street] sleeper OR pavement [dweller] OR floating)". The first wave of the literature search resulted in 459 unique articles, of which co-author DRT screened all titles and abstracts. Of 72 potentially eligible articles from LMICs, DRT reviewed the full-text, and kept five which

## Khomas

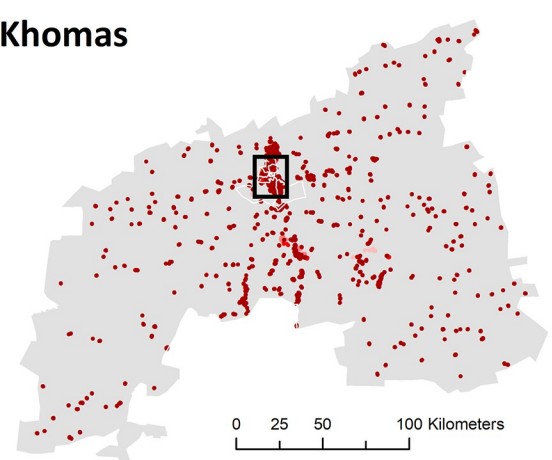

| Households | 2016 | 2011 | 2006 | 2001 |
|---|---|---|---|---|
| Tobias Hainyeko | 12,756 | 12,428 | 10,486 | 8,872 |
| Katutura Central | 5,182 | 5,096 | 4,948 | 4,072 |
| Katutura East | 3,824 | 3,756 | 3,659 | 3,165 |
| Khomasdal | 11,684 | 10,471 | 7,302 | 5,770 |
| Soweto | 3,470 | 3,377 | 3,167 | 2,553 |
| Samora Machel | 16,718 | 13,250 | 7,573 | 6,598 |
| Windhoek East | 7,532 | 7,089 | 6,451 | 5,620 |
| Windhoek Rural | 7,256 | 6,330 | 5,415 | 4,961 |
| Windhoek West | 13,947 | 13,837 | 12,590 | 9,991 |
| Moses //Garoeb | 15,298 | 13,804 | 10,315 | 6,978 |
| **Khomas** | **97,667** | **89,438** | **71,906** | **58,580** |

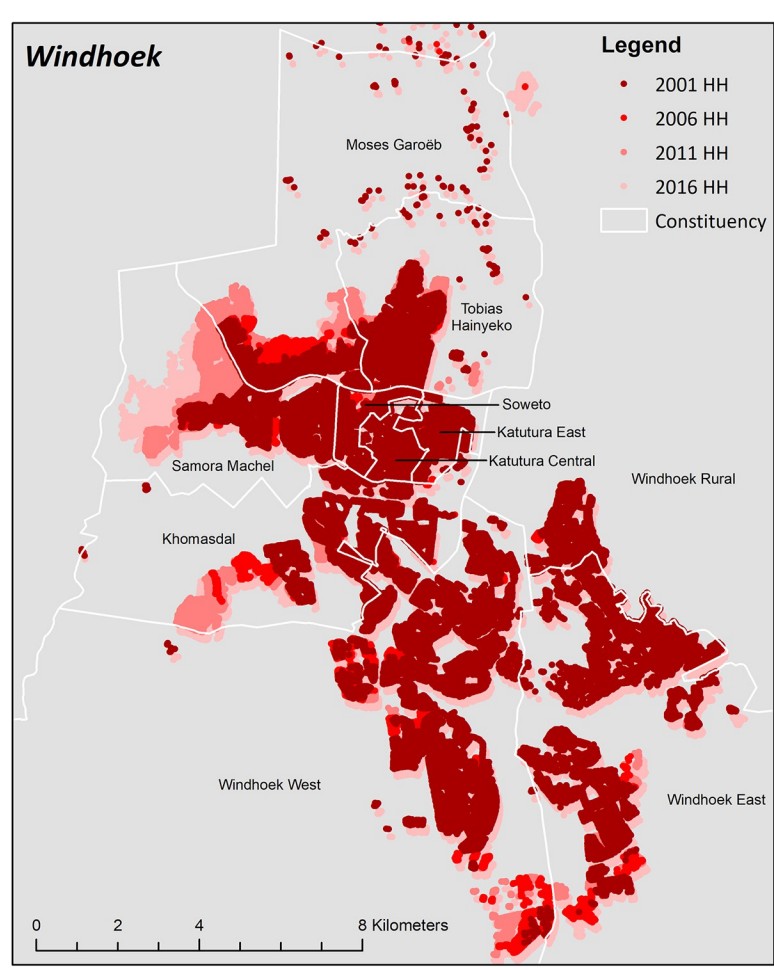

**Fig 3. Household point locations in Khomas, Namibia by presence in 2016, 2011, 2006, and 2001.** Sources: Constituency boundaries publically available from https://gadm.org/. Synthetic population latitude-longitude coordinates available in Supplement 2.

reported a census under-count. In a second wave, we used Google Scholar to identify the top 20 "cited by" and top 20 "related" articles for each of the five articles identified in the first wave. The second wave resulted in 334 unique articles, of which 49 had potentially relevant titles or abstracts. After a full-text review of these articles, we found eight additional reported census under-counts. Together, census under-counts in LMICs were collated from 10 PESs [45–54], and 13 articles [10,55–66] (Fig 4). The average census under-counts were 46% in urban-slum populations, 6% in urban-non-slum populations, and 7% in rural populations (Table 2, see Supplement 3 for details).

Based on these findings, we simulated three levels of census inaccuracy: low inaccuracy was considered to be missing 2% of rural and urban-non-slum households, and 10% of urban-slum households; medium inaccuracy was considered to be missing 5% of rural and urban-non-slum households, and 30% of urban-slum households; and finally, high inaccuracy was classified as missing 10% of rural and urban-non-slum households, and 60% of urban-slum households (Table 1). We applied the inaccuracy rates at random within rural, urban-slum, and urban-non-slum households such that there was no spatial pattern inherent to the simulated under-counts. This exercise resulted in one "true" and 15 outdated-inaccurate simulated

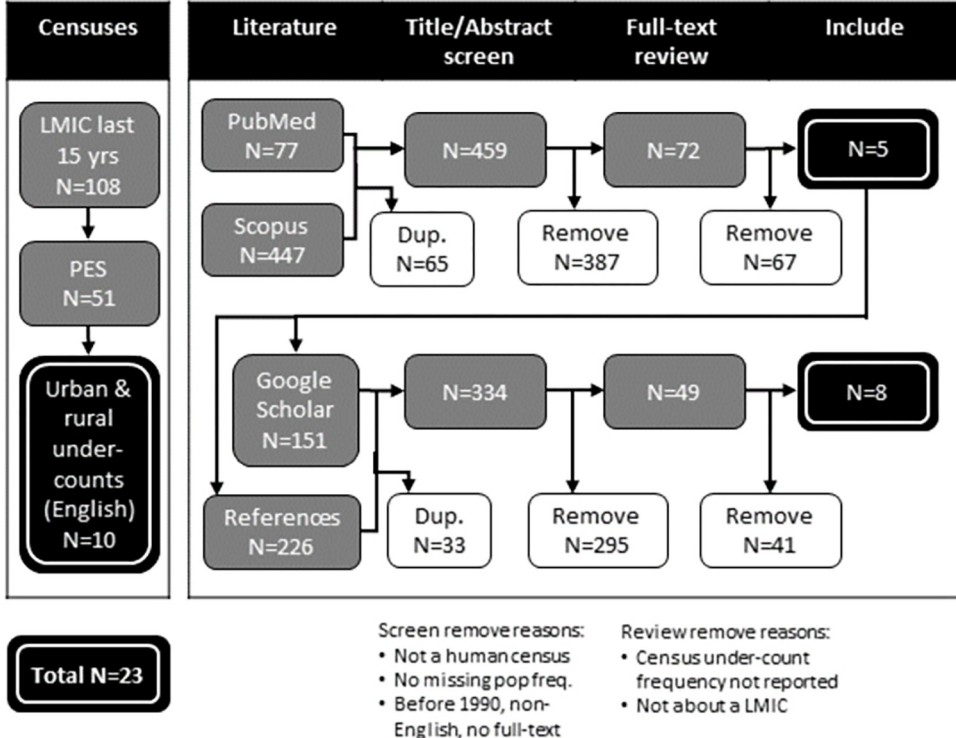

**Fig 4. Search terms and process used in the census under-count literature review.**

populations which we used to generate realistic gridded population datasets that reflect typical gridded population estimates currently available across LMICs (Table 2).

### Simulating realistic gridded population datasets

To simulate realistic gridded population datasets, we aggregated each of the simulated household populations to EA or constituency (second-level administrative unit) boundaries, and applied the WorldPop-Global-Unconstrained modelling technique (for a total of 32 datasets). We applied the WorldPop-Global-Unconstrained model in three phases as described in WorldPop's method publication [22] (Fig 5, Table 3).

1. In the first phase (A), a non-parametric Random Forest ensemble machine-learning algorithm grows a "forest" of decision trees for each input unit (EA or constituency) [67]. Each Random Forest tree is a model of the potential relationships between multiple auxiliary covariates and census population counts. In the Random Forest modelling workflow, this is where model uncertainty is calculated–at the scale of the input population areal unit.

**Table 1. Range and average percent of population missing from LMIC censuses based on literature review.**

| Location | Literature review findings | | | Simulated inaccuracy | | |
|---|---|---|---|---|---|---|
| | Minimum | Average | Maximum | Low | Medium | High |
| Urban-slum | 5% | 46% | 100% | 10% | 30% | 60% |
| Urban-non-slum | 2% | 6% | 15% | 2% | 5% | 10% |
| Rural | 2% | 7% | 13% | 2% | 5% | 10% |

**Table 2. Number of households simulated in the "true" synthetic population and 15 realistic scenarios of census outdatedness and inaccuracy, by household type.**

| Year | No inaccuracy | Low inaccuracy | Medium inaccuracy | High inaccuracy |
|---|---|---|---|---|
| **2016 (current)** | | | | |
| Urban slum | 35,001 | 31,500 | 24,500 | 14,000 |
| Urban non-slum | 57,843 | 56,677 | 54,942 | 52,073 |
| Rural | 4,823 | 4,735 | 4,590 | 4,326 |
| **2011 (5 years old)** | | | | |
| Urban slum | 28,583 | 25,724 | 20,008 | 11,433 |
| Urban non-slum | 55,680 | 54,566 | 52,895 | 50,122 |
| Rural | 5,175 | 5,071 | 4,917 | 4,647 |
| **2006 (10 years old)** | | | | |
| Urban slum | 18,018 | 16,216 | 12,612 | 7,207 |
| Urban non-slum | 49,742 | 48,747 | 47,258 | 44,769 |
| Rural | 4,146 | 4,063 | 3,935 | 3,730 |
| **2001 (15 years old)** | | | | |
| Urban slum | 13,149 | 11,834 | 9,204 | 5,259 |
| Urban non-slum | 41,700 | 40,866 | 39,612 | 37,514 |
| Rural | 3,731 | 3,656 | 3,547 | 3,373 |

Low inaccuracy: missing 2% rural and urban-non-slum households, and 10% urban-slum households. Medium inaccuracy: missing 5% rural and urban-non-slum households, and 30% urban-slum households. High inaccuracy: missing 10% rural and urban-non-slum households, and 60% urban-slum households.

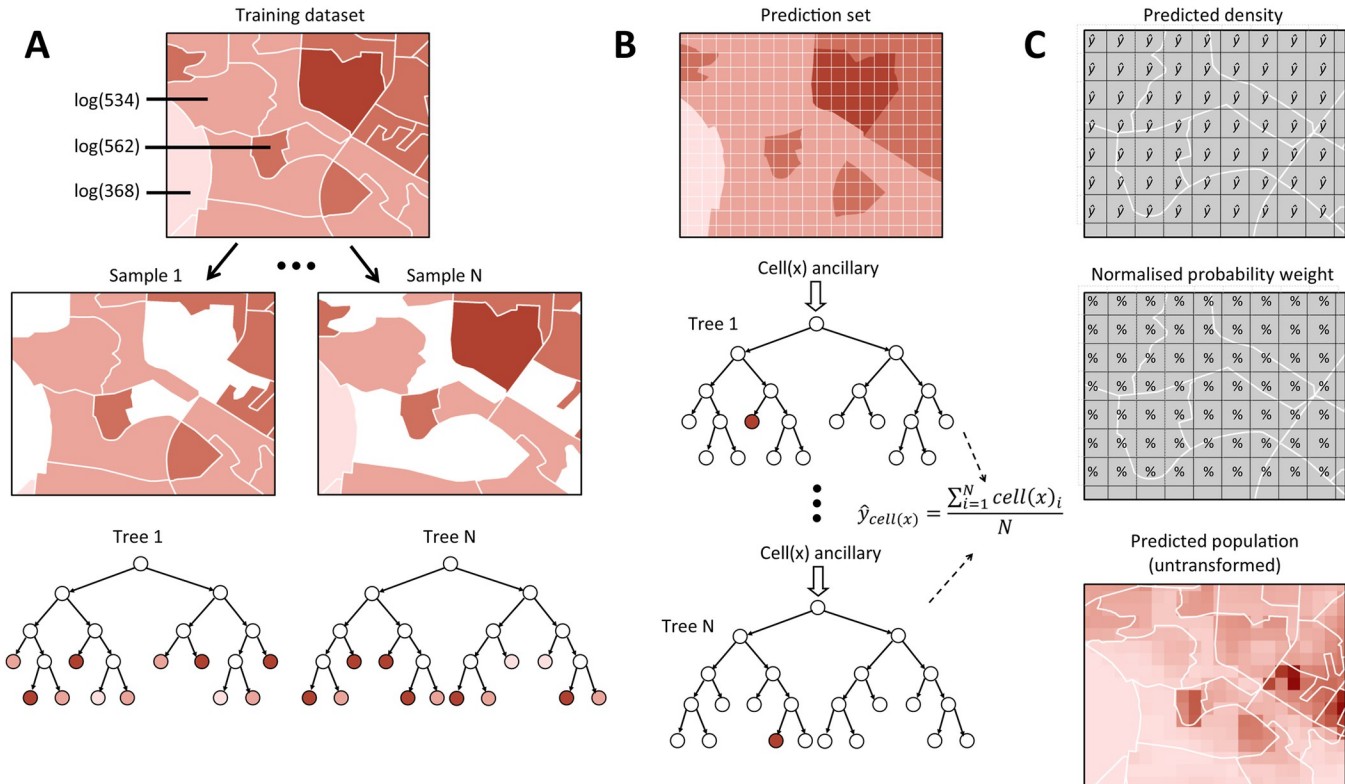

**Fig 5. Overview of WorldPop-Global random forest modelling workflow.** (A) Each decision tree in the ensemble is built upon a random bootstrap sample of the log-transformed population and ancillary data by administrative unit. (B) Population density prediction for each cell $y_{cell(x)}$ is based on an average of the individual trees. (C) Predicted cell densities are normalized by administrative unit and used to dasymetrically disaggregate log-transformed administrative unit population, then transformed to predict population per cell.

$$\hat{y}_{cell(x)} = \frac{\sum_{i=1}^{N} cell(x)_i}{N}$$

**Table 3. Covariate data sources for Random Forest gridded population estimates.**

| Name | Description (Year) | Original scale | Original source |
|---|---|---|---|
| cov_road | Distance to OSM major roads (2016) | Vector, <30 m | OpenStreetMap [68] |
| cov_intsec | Distance to OSM major road intersections (2016) | Vector, <30 m | OpenStreetMap [68] |
| cov_waterw | Distance to OSM major waterways (2016) | Vector, <30 m | OpenStreetMap [68] |
| cov_wdpa | Distance to IUCN nature reserve (2000–17) | 30" (~900 m) | UNEP-WCMS & IUCN [69] |
| cov_viirs | Resampled VIIRS night-time lights (2012–2016) | 30" (~900 m) | NOAA [70] |
| cov_dmsp | Resampled DMSP-OLS night-time lights (2011) | 30" (~900 m) | NOAA & Zhang, et al. [71,72] |
| cov_tt50k | Resampled travel time to cities of 50,000+ (2000) | 30" (~900 m) | Weiss, et al. [73] |
| cov_001 | Distance to cultivated areas (2015) | 9" (~300 m) | ESA CCI–LC [74] |
| cov_040 | Distance to woody areas (2015) | 9" (~300 m) | ESA CCI–LC [74] |
| cov_130 | Distance to cultivated areas (2015) | 9" (~300 m) | ESA CCI–LC [74] |
| cov_140 | Distance to herbaceous areas (2015) | 9" (~300 m) | ESA CCI–LC [74] |
| cov_150 | Distance to sparse vegetation areas (2015) | 9" (~300 m) | ESA CCI–LC [74] |
| cov_160 | Distance to aquatic vegetation areas (2015) | 9" (~300 m) | ESA CCI–LC [74] |
| cov_190 | Distance to urban areas (2015) | 9" (~300 m) | ESA CCI–LC [74] |
| cov_200 | Distance to bare areas (2015) | 9" (~300 m) | ESA CCI–LC [74] |
| cov_cciwat | Distance to ESA-CCI-LC inland waterbodies (2000–12) | 4.5" (~150 m) | ESA CCI [75] |
| cov_slope | SRTM-based slope (2000) | 3" (~90 m) | de Ferranti [76,77] |
| cov_topo | SRTM-based elevation (2000) | 3" (~90 m) | de Ferranti [76,77] |
| cov_coast | Distance to open-water coastline (2000–20) | 3" (~90 m) | CIESIN [78] |
| cov_ghsl | Distance to urban area (2012) | 1.26" (~38 m) | Pesaresi, et al. [79] |
| cov_guf | Distance to settlement built-up areas (2012) | 2.8" (~84 m) | DLR EOC [80] |
| cov_bsgme | Distance to built settlement expansion (2016) | 3" (~90 m) | Nieves, et al. [81] |
| cov_prec | Average total annual precipitation (1970–2000) | 30" (~900 m) | Fick and Hijmans [82] |
| cov_temp | Average annual temperature (1970–2000) | 30" (~900 m) | Fick and Hijmans [82] |

OSM: OpenStreetMap; VIIRS: Visible Infrared Imaging Radiometer Suite; DMSP-OLS: Defence Meteorological Satellite Program Operational Linescan System; ESA-CCI-LC: European Space Agency Climate Change Initiative Land Cover; UNEP-WSMS: UN Environment Programme World Conservation Monitoring Centre; IUCN: International Union for Conservation of Nature; NOAA: US National Oceanic and Atmospheric Administration; CIESIN: Center for International Earth Science Information Network; DLR EOC: German Aerospace Center Earth Observation Center.

2. In the second phase (B), all of the covariates are prepared in 100x100m cells. In this phase, the split values of each classification tree developed in phase A are used to parameterize corresponding regression models to predict population density within 100x100m cells [22]. For each cell, the predicted population values from all regression models are averaged to make a single population estimate, though these population estimates are not pycnophylactic, meaning that estimates in cells do not necessarily sum to the original areal unit population.

3. Thus the WorldPop-Global-Unconstrained workflow involves a third phase (C) outside of the Random Forest model to normalize cell-level predicted population densities to preserve census input population counts [22].

## Analysing cell-level accuracy

To empirically measure cell-level accuracy of the 32 gridded population datasets, we compared each cell-level estimate against the "true" synthetic point-level 2016 population count in that cell, then calculated root mean square error (RMSE), a measure of error magnitude that penalises large errors. This was performed on 100x100m cells, and then estimated cell population

counts were aggregated and assessed for accuracy at 200x200m, 300x300m, and so on up to 1x1km. This was to test a common assumption that large model errors at fine geographic scale are "smoothed out" and become less severe as population estimates are aggregated across larger zones. To compare RMSE across cells of different geographic sizes, we normalised the statistic by average population (Eq 1) and by area (Eq 2). The former represents RMSE of population counts expressed as a portion of the population [83], while the latter represents RMSE of population density per hectare (100x100m unit) [84]. We evaluated RMSE in urban-slum, urban-non-slum, and rural cells separately. In the calculation of RMSE, $y_i$ is the "true" synthetic population count in cell $i$, $\hat{y}_i$ is the gridded population estimate in cell $i$, $D_i$ is the "true" synthetic population density per hectare, $\hat{D}_i$ is the estimated population density per hectare, and $n$ is the number of grid cells.

$$Pop - adj\ RMSE\ =\ \sqrt{\frac{\sum_{i=1}^{n}(\hat{y}_i - y_i)^2}{n}}\ \div\ \frac{\sum_{i=1}^{n}(y_i)}{n} \qquad 1$$

$$Area - adj\ RMSE\ =\ \sqrt{\frac{\sum_{i=1}^{n}(\hat{D}_i - D_i)^2}{n}} \qquad 2$$

To better understand the mechanics of the WorldPop-Global-Unconstrained model and workflow, we calculated bias, a measure of error direction and magnitude. This metric was especially useful for the two gridded population datasets derived from "true" synthetic population counts because any inaccuracies would be related to the model and covariate datasets alone; and not inaccuracies in the input population counts. Bias (Eq 3) reveals to what extent cell-level estimates are systematically under- or over-estimated, and reflects over/under-counts in cells of different sizes that a user might encounter in the field. Relative bias (Eq 4) refers to bias normalised by the average synthetic population which enables comparisons across grid scales. As above, bias and relative bias were assessed in 100x100m cells as well as cell sizes that ranged up to 1x1km, and separately in urban versus rural areas.

$$Bias\ =\ \frac{\sum_{i=1}^{n}(\hat{y}_i - y_i)}{n} \qquad 3$$

$$Relative\ bias\ =\ \frac{\sum_{i=1}^{n}(\hat{y}_i - y_i)}{n}\ \Big/\ \frac{\sum_{i=1}^{n}(y_i)}{n} \qquad 4$$

To assess the degree to which non-zero population estimates in the WorldPop-Global-Unconstrained dataset resulted in misalocation of population, a third statistic was calculated counting the entire modelled population in Khomas that was misallocated to cells which were unsettled according to the "true" synthetic population. For all statistics, we excluded gridded population cell-level estimates of less than 1 person to avoid millions of near-zero cell-level estimates in unsettled areas of Khomas (located outside of Windhoek) from dominating the accuracy assessments.

## Results

Neither measure of RMSE differed substantially across the simulated outdated-inaccurate census scenarios (Figs 6 and 7). Furthermore, errors only slightly decreased when the input data were aggregated to EA (finer) rather than constituency (coarser) (Figs 6 and 7). The major driver of RMSE in cells was urban versus rural location, with further difference between urban-slum and urban-non-slum. In urban cells, population-adjusted RMSE was

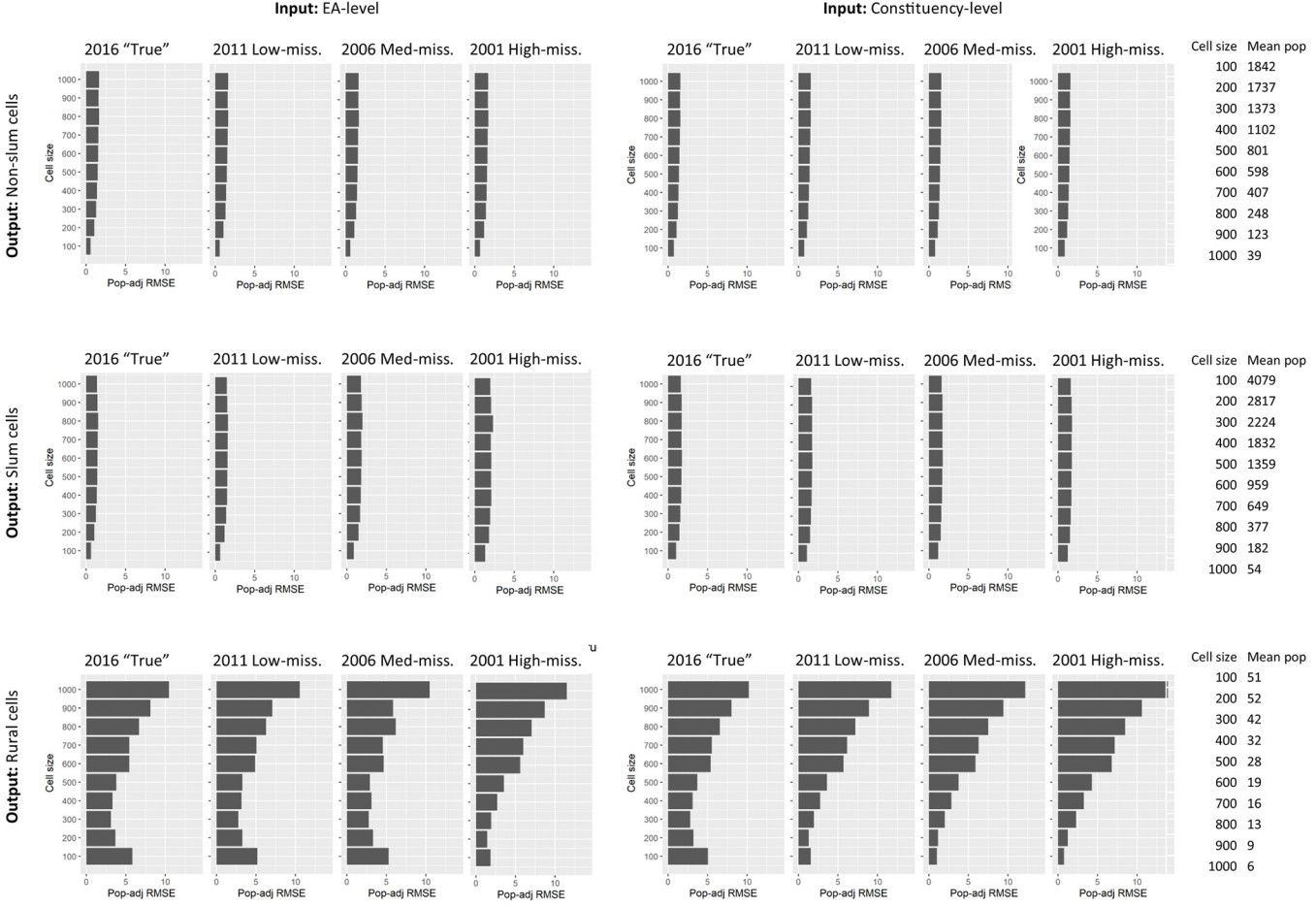

**Fig 6. Population-adjusted root mean square error (RMSE) according to input population aggregation, a selection of scenarios, and cell size.**

substantially smaller than rural cells (Fig 6), but much larger per hectare due to larger population numbers (Fig 7). In urban areas, RMSE per hectare was lowest in 100x100m cells (slum range: 32–72, non-slum range: 21–33), while in rural areas, RMSE per hectare was lowest in cells 300x300m to 500x500m (rural range: 2–54) (Fig 7). Results for select scenarios are presented in Fig 6 ranging from the synthetic "true" 2016 population to the most outdated (2001) and inaccurate (missing 10% to 60%) population, though tables of all results are provided in Supplement 4.

Assessment of bias in the two gridded population datasets that were derived from synthetic "true" 2016 population counts revealed systematic and substantial under-estimates of populations in urban-slum and urban-non-slum cells due to the aggregation-level of the input population data and modelling approach, and not inaccuracies in the input data (Tables 4 and 5). For example, the average 300x300m urban-slum cell under-estimated the population by more than 350 people (EA-level input) up to 500 people per cell (constituency-level input). For comparison, the average 300x300m non-slum cell was under-estimated by 165 people (constituency-level input) to 187 people (EA-level input), while the average rural cell of the same size was over-estimated by 3 people (constituency-level input) to 14 people (EA-level input) (Table 4). When adjusted for population, the results indicate that for every person estimated in

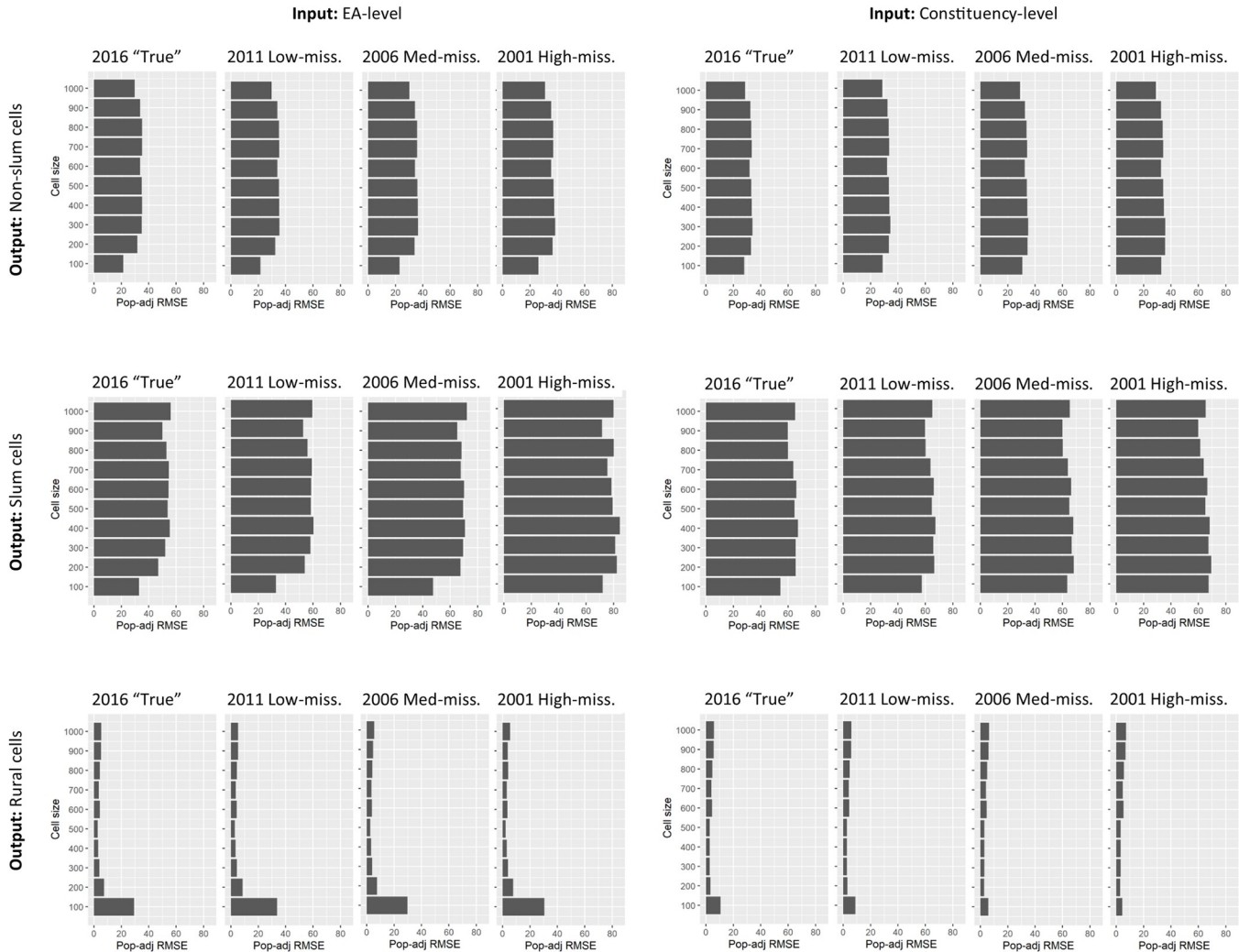

**Fig 7. Population density root mean square error (RMSE) per hectare according to input population aggregation, a selection of scenarios, and cell size.**

an urban non-slum cell, 0.5 to 1 person is omitted; and for every person estimated in an urban slum cell, 0.75 to 1.5 people are omitted (Table 5).

Table 6 summarises the percent of the estimated population misallocated to "truly" unsettled cells according to the synthetic population. For this analysis, no cells in the estimated population were excluded. Roughly 20% (EA-level input) or 10% (constituency-level input) of the population was misallocated to unsettled 100x100m cells (Table 6). However, as cells were aggregated, the percent of misallocated population dropped precipitously. For example, at 300x300m, approximately 2% (EA-level input) or 1% (constituency-level input) of Khomas's population was misallocated to unsettled cells. This indicates that most of the population was disaggregated to unsettled cells within, or near to, settlements. The rates of misallocation were similar when cells with less than one person were excluded (not reported).

**Table 4. Bias in gridded population estimates derived from "true" synthetic population counts, by output grid cell size and urban/rural location (in cells > = 1 estimated person).**

| Cell size | EA-level input | | | Constituency-level input | | |
|---|---|---|---|---|---|---|
| | **Non-slum** | **Slum** | **Rural** | **Non-slum** | **Slum** | **Rural** |
| 100 | 0 | 0 | 20 | -4 | -34 | 7 |
| 200 | -71 | -135 | 18 | -64 | -212 | 6 |
| 300 | -187 | -353 | 14 | -165 | -498 | 3 |
| 400 | -346 | -678 | 8 | -303 | -929 | -1 |
| 500 | -549 | -1029 | 3 | -483 | -1401 | -8 |
| 600 | -769 | -1480 | -22 | -672 | -2080 | -34 |
| 700 | -1094 | -2114 | -33 | -981 | -2747 | -51 |
| 800 | -1410 | -2692 | -72 | -1247 | -3359 | -90 |
| 900 | -1728 | -3215 | -126 | -1576 | -4437 | -152 |
| 1000 | -1928 | -4453 | -126 | -1770 | -5834 | -167 |

## Discussion

This is among the first accuracy assessments of a top-down gridded population model at the grid-cell level, and the first that we know of in a LMIC setting. By developing a simulated realistic population and several scenarios of the population with realistic levels of outdatedness and inaccuracy, we were able to evaluate the accuracy of a gridded population model, as well as assess the impact of outdated-inaccurate census inputs on estimates. In this paper, we evaluated just one of several gridded population models–WorldPop-Global-Unconstrained. We also only analysed one simulated population and focused on the particular setting of Khomas, Namibia, so the results do no necessarily generalize to other cities or datasets. In this specific analysis, cell-level inaccuracies between urban versus rural areas dominated the results.

In practical terms, this massive difference between urban versus rural accuracy means that urban development indicators calculated with a WorldPop-Global-Unconstrained dataset at fine scale (e.g., neighbourhood) would likely be incorrect, and could lead to poorly informed decisions. For example, an underestimate of the number of people living in a neighbourhood could overestimate both vaccination coverage and disease infection rates. Contrary to what some might assume, there was limited evidence in this study that outdated or inaccurate census data played a major role in cell-level inaccuracy of gridded population estimates. Instead, we address three other potential sources of the cell-level inaccuracies observed.

**Table 5. Population-adjusted bias in gridded population estimates derived from "true" synthetic population counts, by output grid cell size and urban/rural location (in cells > = 1 estimated person).**

| Cell size | EA-level input | | | Constituency-level input | | |
|---|---|---|---|---|---|---|
| | **Non-slum** | **Slum** | **Rural** | **Non-slum** | **Slum** | **Rural** |
| 100 | 0.00 | 0.00 | 3.36 | -0.10 | -0.64 | 1.28 |
| 200 | -0.58 | -0.74 | 1.90 | -0.52 | -1.16 | 0.67 |
| 300 | -0.76 | -0.94 | 1.07 | -0.67 | -1.32 | 0.23 |
| 400 | -0.85 | -1.04 | 0.52 | -0.74 | -1.43 | -0.08 |
| 500 | -0.92 | -1.07 | 0.15 | -0.81 | -1.46 | -0.40 |
| 600 | -0.96 | -1.09 | -0.77 | -0.84 | -1.53 | -1.21 |
| 700 | -0.99 | -1.15 | -1.02 | -0.89 | -1.50 | -1.59 |
| 800 | -1.03 | -1.21 | -1.70 | -0.91 | -1.51 | -2.12 |
| 900 | -1.00 | -1.14 | -2.41 | -0.91 | -1.58 | -2.92 |
| 1000 | -1.05 | -1.09 | -2.45 | -0.96 | -1.43 | -3.25 |

The first issue is specific to the WorldPop-Global-Unconstrained modelling approach. In this approach, input administrative units with zero population are excluded and the remaining population counts are log-transformed before inclusion in a Random Forest model. While this procedure ensures that population counts are normally distributed during modelling, it also means that unpopulated cells are assigned a very small fraction of a person [22]. A possible concern is that non-zero population estimates across millions of unsettled cells could result in a sizable portion of the population being misallocated. Our analysis of misallocation, however, indicates that this phenomenon played only a minor role in cell-level inaccuracies, if at all. Table 6 demonstrates that even in this context of vast unsettled areas, only a small portion of Khomas' population was misallocated to cells far from actual settlements. Nearly all of the population was estimated to be in cells within 200 to 300 metres of the "true" synthetic population.

Most global gridded population producers constrain estimates to settled cells as defined with a settlement layer (e.g. LandScan [24,85], GHP-POP [19,20], HRSL [21], GRID3 [28,86], WPE [26]). Until recently, these settlement layers tended to be relatively coarse (e.g. GHS-BUILT 1x1km [87]) and/or had a tendency to omit areas with few sparse buildings (e.g. GUF [80]) which could under-estimate the population in rural areas and over-estimate the population in urban areas. However, new free very high resolution Sentinel-2 imagery, and major leaps in computing power for extracting building footprints and other features from imagery, have enabled development of several new detailed settlement layers in the last few years (e.g., GHS-BUILT-S2 [88], Maxar/Ecopia [89]). Recently, WorldPop-Global produced a constrained global gridded population estimate for 2020 that uses the same input population and covariate datasets as its unconstrained model plus several building footprint metrics (in Africa), and then masks all 100x100m cells without building footprints (in Africa) or built settlement (rest of the world) [35], eliminating the issue of non-zero population estimates in unsettled cells.

The second potential source of inaccuracy relates to covariate resolution and the relationship of covariates with population density. This issue seems to have contributed more substantially to errors in this analysis, particularly within the city of Windhoek. A number of the Random Forest model covariates, such a land cover type and night-time lights, had an original resolution substantially coarser than 100x100m which could have resulted in a "halo" effect around settlements, causing populations to be disaggregated to cells near a settlement, but not directly over it. Table 5 provides evidence of this; the accuracy of the estimated population distribution, and correct allocation of population to settled cells, both performed well when the estimated population was aggregated to 300x300m or larger. Other covariates, such as distance

**Table 6. Percent of the overall population that is misallocated to unsettled cells (no exclusion), by aggregation level of the input data and output grid cell size.**

| Grid cell size (m²) | EA-Level Input | Constituency-Level Input |
|---|---|---|
| 100 | 20.8% | 12.5% |
| 200 | 5.0% | 2.6% |
| 300 | 2.2% | 1.0% |
| 400 | 1.3% | 0.5% |
| 500 | 0.8% | 0.3% |
| 600 | 0.6% | 0.2% |
| 700 | 0.4% | 0.1% |
| 800 | 0.3% | 0.1% |
| 900 | 0.3% | 0.1% |
| 1000 | 0.2% | 0.1% |

to roads and intersection locations were available at very fine spatial resolution and thus were precise at the 100x100m scale. Although they are good indicators of a settlement, they are not necessarily good indicators of higher or lower population density within a settlement. The lack of fine-scale covariates associated with population density within cities and towns likely explains a portion of the cell-level error observed in Khomas's urban population. Other issues that might further decrease local spatial accuracy are temporal miss-match of covariates [16] and covariate spatial autocorrelation [90]. With the recent release of several building footprint datasets (e.g., Maxar/Ecopia in most of Africa [89], Bing in Tanzania and Uganda [91]), several new covariate layers have been created by the WorldPop team including number of buildings and total area of buildings in 100x100m cells [92]. Building footprints are likely associated with population density within settlements and have a finer spatial resolution than 100x100m, making it a potentially powerful covariate to differentiate lower and higher population density within urban areas in any gridded population model. The WorldPop team, among other gridded population producers, is currently working to test and incorporate building footprint datasets into gridded population models.

The third potential source of cell-level inaccuracies is use of average population densities from large administrative units to estimate population densities in much smaller grid cells. This is known as the ecological fallacy [93], and probably played the largest role in cell-level inaccuracies, especially within urban areas. Population densities are used by the Random Forest model to establish relationships between covariates and population *density* (total population divided by total area), not population totals. Even with perfect covariates and exclusion of unsettled areas, this would mean that cells with high "true" synthetic population counts are likely to be severely underestimated because the geographic size of input population units are larger (and population densities are smaller) than the output grid cells. When this happens, population counts that are not allocated to the densest cells will instead be allocated to other less dense cells in the same input areal unit. Tables 4 and 5 provide strong evidence of this issue with the population in urban cells, especially urban-slum cells, systematically underestimated regardless of cell size.

Although these results apply only to the WorldPop-Global-Unconstrained model, we can speculate about how these results might apply to other gridded population datasets. Most top-down gridded population datasets use average population densities from large input areal units in some way to populate smaller grid cells, and are thus likely subject to similar errors linked with the ecological fallacy. The High Resolution Settlement Layer (HRSL), for example, uses uniform areal disaggregation of the population from input units (e.g., EA) to 30x30m grid cells which contain a building footprint [21], and the Global Human Settlement GHS-POP dataset takes a similar approach disaggregating input populations into 250x250m cells that are classified as settled [19,20]. Gridded Population of the World (GPWv4) is likely even less accurate at the cell-level because the population from each input unit (e.g., EA) are smoothed across all cells in that unit, including unsettled cells [17]. Gridded population datasets based on complex models with variable disaggregation from units to grid cells, such as LandScan [24] and World Population Estimates (WPE) [26], are instead subject to the second limitation described above because, like WorldPop-Global-Unconstrained, they lack high-resolution model covariates (e.g., building density) to accurately differentiate population density within settled cells.

This analysis reinforces findings of other studies which find that currently available gridded population products tend to underestimate populations in urban areas [94–96], especially in higher-density poorer neighbourhoods [97]. For example, Tuholske and colleagues (2021) compared five gridded population products to estimate the proportion of population affected by natural disasters (SDG 11.5) in three regions where disasters had occurred, and found that

1x1km population estimates varied widely among data products, and reflected anywhere from 20% to 80% of the total UN estimated population in each region. Furthermore, they found that WorldPop-Global-Unconstrained generally performed better than un-modelled products (e.g., GPW), but not as well as products that constrained estimates to settled cells (e.g., GHS-POP) [94]. In a separate comparison of nine gridded population estimates of Kenyan and Nigerian slum populations (SDG 11.1.1) where field counts were available for reference, the estimated population in each slum varied widely and WorldPop-Global-Unconstrained estimates reflected just 11% of the overall slum population while the best performing data product (HRSL) estimated just 34% of all slum dwellers [97]. A key take-away from gridded population comparison studies is that fine-scale accuracy across data products varies substantially depending on location, potentially leading to different conclusions and decisions (e.g., about the humanitarian need or health care burden) depending on the gridded population dataset used for analysis. Furthermore, these studies underscore the need to understand fine-scale accuracy across gridded population datasets and locations to inform improvements to the underlying modelling methods and inputs.

Our analysis of a simulated population offers a methodological approach that can be replicated in other settings to evaluate the accuracy of any gridded population dataset at the cell-level. This analysis also points toward two solutions–use of building footprint covariates and finer-scale training data–that stand to improve cell-level accuracy of gridded population datasets derived from complex models, including all WorldPop-Global datasets as well as Land-Scan [24,25], WPE [26], and GRID3 [28,86]. Other techniques would be needed to improve the accuracy of gridded population datasets that do not vary (weight) population densities within areal units based on auxiliary information (e.g., HRSL [21], GHS-POP [19,20], GPW [17,18]).

Our first suggestion to improve WorldPop-Global datasets is to incorporate finer-scale training data into models to overcome the problem of larger areal-unit average values being used in smaller grid cells. In cases where the input areal units are geographically large, World-Pop-Global-Unconstrained (and Constrained) models incorporate training data from a neighbouring country that has finer-scale input population counts [22]. Our analysis showed, however, that even when relatively small geographic units (census EAs) were used as the input population area unit, urban slum and non-slum cell-level errors were still substantial, and cell-level accuracy with EA-level input was only marginally improved compared to constituency-level input (Fig 7). This suggests that finer-scale training data (e.g., closer to 100x100m) should be incorporated during the model training phase, particularly from high-density urban areas, to ensure that the WorldPop Random Forest model contains sufficiently large population density values to assign to urban cells. Fine-scale training datasets might come from existing household survey enumerations (e.g., World Bank Living Standards Measurement Surveys), or slum community profiles such as those published on the Know Your City Campaign website [98]. Even if fine-scale densities are only available for a small sample of locations, they would provide the model with more accurate maximum population values at the scale of 100x100m during model training.

The second potential solution is to incorporate more spatially detailed datasets into models which correlate with variations in population density. This analysis of WorldPop-Global-Unconstrained data raises broader questions about the cell-level accuracy of all gridded population estimates in urban areas, especially the densest parts of cities such as in slums, informal settlements, and neighbourhoods with high-rise apartment buildings [99–101]. New datasets derived from very high resolution satellite imagery, in particular building footprints, are a promising new covariate to reduce the "halo" effect of populations misallocated nearby, but not directly over, the highest density cells. More work will be needed to improve building

footprint datasets by distinguishing residential and non-residential buildings to avoid population being misallocated to business districts, factories, universities, airports, and other non-residential cells [102,103].

## Conclusions

Global gridded population data initiatives aim to fill a gap in available disaggregated and current population counts to ensure that everyone is counted and that all needs are met in development initiatives. However, many gridded population datasets are not evaluated for accuracy at fine spatial scale. This analysis of one simulated population in one setting revealed substantial and systematic under-estimation of population in slums. Further analyses of other gridded population datasets are needed across diverse settings. However, if severe under-estimates in slums and other high-density urban areas are widespread, this means that gridded population datasets might unintentionally reinforce marginalisation of the urban poorest by omitting them from maps and population counts. We offer two suggestions to address this challenge: inclusion of finer-scale training data from household survey listings or "slum" enumerations, and the addition of new building footprints data as model covariates. Given the increased use of gridded population datasets for monitoring health and development outcomes in small areas, it is imperative that gridded population datasets area assessed for cell-level accuracy and are improved where possible.

## Supporting information

**S1 Table. Percent of population missing from LMIC censuses by source.**
(DOCX)

**S2 Table. Root Mean Square Error (RMSE) statistics for all scenarios.**
(DOCX)

**S1 File. Simulating a population in Khomas, Namibia.**
(PDF)

**S2 File. Simulated population in Khomas, Namibia.**
(CSV)

## Acknowledgments

We would like to thank Drs. Angela Luna Hernandez and Ryan Engstrom for their feedback on an earlier version of this work.

## Author Contributions

**Conceptualization:** Dana R. Thomson.

**Data curation:** Dana R. Thomson.

**Formal analysis:** Dana R. Thomson.

**Methodology:** Dana R. Thomson, Douglas R. Leasure, Tomas Bird, Nikos Tzavidis, Andrew J. Tatem.

**Supervision:** Douglas R. Leasure, Tomas Bird, Nikos Tzavidis, Andrew J. Tatem.

**Visualization:** Dana R. Thomson.

**Writing – original draft:** Dana R. Thomson.

**Writing – review & editing:** Dana R. Thomson, Douglas R. Leasure, Tomas Bird, Nikos Tzavidis, Andrew J. Tatem.

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
