## [Decision Letter · Decision Letter 0]

28 Jul 2021

PONE-D-21-16343

How accurate are WorldPop-Global-Unconstrained gridded population data at the cell-level?: A simulation analysis in urban Namibia

PLOS ONE

Dear Dr. Thomson,

Thank you for submitting your manuscript to PLOS ONE. After careful consideration, we feel that it has merit but does not fully meet PLOS ONE’s publication criteria as it currently stands. Therefore, we invite you to submit a revised version of the manuscript that addresses the points raised during the review process.

We look forward to receiving your revised manuscript.

Kind regards,

Krishna Prasad Vadrevu, Ph.D

Academic Editor

PLOS ONE

2.Please note that in order to use the direct billing option the corresponding author must be affiliated with the chosen institute. Please either amend your manuscript to change the affiliation or corresponding author, or email us at plosone@plos.org with a request to remove this option.

3. We note that Figures 1 &3  in your submission contain [map/satellite] images which may be copyrighted. All PLOS content is published under the Creative Commons Attribution License (CC BY 4.0), which means that the manuscript, images, and Supporting Information files will be freely available online, and any third party is permitted to access, download, copy, distribute, and use these materials in any way, even commercially, with proper attribution. For these reasons, we cannot publish previously copyrighted maps or satellite images created using proprietary data, such as Google software (Google Maps, Street View, and Earth). For more information, see our copyright guidelines: http://journals.plos.org/plosone/s/licenses-and-copyright.

a. You may seek permission from the original copyright holder of Figures 1 & 3 to publish the content specifically under the CC BY 4.0 license. 

Additional Editor Comments (if provided):

Dear Authors,

First, thank you for submitting the manuscript to PLOS ONE. We have received reviews from two different experts with one major and another minor revision. Based on the suggestions, we ask you to submit a revised manuscript. Specifically, please see that the revised version includes clarification on the a). possibilities of extrapolating the results to other regions; b). the usefulness of the research for applications; c). Use of RMSE versus MAE while evaluating the gridded population datasets; d). Figures improvement, etc.

We look forward to a revised version.

Best,

Krishn

Reviewers' comments:

Reviewer's Responses to Questions

**Comments to the Author**

1. Is the manuscript technically sound, and do the data support the conclusions?

Reviewer #1: Yes

Reviewer #2: Yes

2. Has the statistical analysis been performed appropriately and rigorously? 

Reviewer #1: Yes

Reviewer #2: Yes

3. Have the authors made all data underlying the findings in their manuscript fully available?

Reviewer #1: No

Reviewer #2: Yes

4. Is the manuscript presented in an intelligible fashion and written in standard English?

Reviewer #1: No

Reviewer #2: Yes

5. Review Comments to the Author

Reviewer #1: I find the topic interesting and the authors show to be experts in the topic. However, I think that the paper is too technical for most potential readers. I provide some simple comments hoping they can help in this regard.

1.The abstract is extremely and unnecessarily long.

2. I find the intro too technical. I recommend leaving technical thing for other sections and devoting more in the intro to tell the reader i) why is the topic relevant for policy debates, ii) what are the main contribution of the current paper?

3. I like the idea of a short and synthetic paper. However, even in a paper of this style, I think framing the paper in the relevant literature is essential. There are several papers on urbanisation, urban density, suburbanisation, etc. worldwide that the paper should cite and relate to. I recommend recent papers in the Journal of Economic Geography, Journal of Development Studies and in the Journal of Urban Economics.

4. Could the text be easier to read, leaving some technicalities for an appendix?

5. Sorry but I find really puzzling the use of “simulated” and “true” in the same sentences over and over in the paper to refer to the same numbers. How can be, at the same time, “simulate” and “true”?!

6. How could we extrapolate the findings for Namibia to other wold regions?

7. Finally, I miss a connection with applied research. For user of data sources like Gridded Population of the World, what does all mean? What are the implications? Alternatives? Etc. I think the authors should make discuss all this, leaving technicalities aside, in the conclusions

Reviewer #2: Review of the manuscript “How accurate are WorldPop-Global-Unconstrained gridded population data at the cell-level?:A simulation analysis in urban Namibia”

As the authors point out, the paper presents a method of evaluating the cell-level accuracy of 32 simulated 100x100m WorldPop-Global-Unconstrained gridded population datasets which reflect realistic scenarios of census (1) outdatedness, (2) inaccuracy, and (3) aggregation in an urban LMIC setting. This topic is very interesting and timely, but the purpose of the article should be described more clearly.

A thorough overview of the literature is included in Introduction, and the quoted items exhaust the proposed topic. In line 123 authors state that they evaluate 32 simulated 100x100m WorldPop-Global-Unconstrained gridded population datasets. The authors should explain why they chose the 32 grid. What was the reason for choosing such a set of gridded population datasets?

The section Methods is well presented and illustrated with figures. Yet, it should be explained why the Root Mean Square Error (RMSE) was selected to evaluate the gridded population dataset. The literature provides ample evidence on the effectiveness and usefulness of the Mean Absolute Error (MAE).

The quality of figures and charts is quite unsatisfactory, and they need to be presented in adequate resolution.

In Figure 1, the black background interferes with map reading. What do white boundaries on the right of the map mean?

The Discussion section is presented in a clear way. Will the studies be continued?

Is the proposed method universal? Can it be used for other research areas?

The Conclusions section reinstates the main findings in an adequate way.

The article meets high scientific quality standards and fits the scope of PLOS ONE. It contributes to the existing knowledge, presenting the topic in an interesting and up-to-date way.

6. PLOS authors have the option to publish the peer review history of their article (what does this mean?). If published, this will include your full peer review and any attached files.

Reviewer #1: No

Reviewer #2: No

---

## [Author Response · Author response to Decision Letter 0]

1 Oct 2021

Please see Response to Reviewers letter.

---

## [Decision Letter · Decision Letter 1]

10 Nov 2021

PONE-D-21-16343R1How accurate are WorldPop-Global-Unconstrained gridded population data at the cell-level?: A simulation analysis in urban NamibiaPLOS ONE

Dear Dr. Thomson,

Thank you for submitting your manuscript to PLOS ONE. After careful consideration, we feel that it has merit but does not fully meet PLOS ONE’s publication criteria as it currently stands. Therefore, we invite you to submit a revised version of the manuscript that addresses the points raised during the review process.

Please elaborate the discussion to include Application users in mind. Also, please see the suggestions on additional literature suggested by one of the reviewers - please refer and cite them as needed.

We look forward to receiving your revised manuscript.

Kind regards,

Krishna Prasad Vadrevu, Ph.D

Academic Editor

PLOS ONE

Journal Requirements:

Reviewers' comments:

Reviewer's Responses to Questions

**Comments to the Author**

1. If the authors have adequately addressed your comments raised in a previous round of review and you feel that this manuscript is now acceptable for publication, you may indicate that here to bypass the “Comments to the Author” section, enter your conflict of interest statement in the “Confidential to Editor” section, and submit your "Accept" recommendation.

Reviewer #1: (No Response)

Reviewer #2: All comments have been addressed

2. Is the manuscript technically sound, and do the data support the conclusions?

Reviewer #1: Yes

Reviewer #2: Yes

3. Has the statistical analysis been performed appropriately and rigorously? 

Reviewer #1: Yes

Reviewer #2: Yes

4. Have the authors made all data underlying the findings in their manuscript fully available?

Reviewer #1: No

Reviewer #2: Yes

5. Is the manuscript presented in an intelligible fashion and written in standard English?

Reviewer #1: Yes

Reviewer #2: Yes

6. Review Comments to the Author

Reviewer #1: I acknowledge the work done in revising the manuscript. I think that the paper has improved significantly.

The Introduction is now much easier to read and it better motivates the paper.

I still find the paper very technical, but I understand that this is the contribution. My concern is that, given its focus and style, the reach of the paper will be limited (see my next comment). In this line, the new intro helps. The discussion had not changed much and could try to be broader in scope.

Related to the above, the literature continuous to be deficient. Most references are technical. For applied “users” of gridded data (rather than researchers “creating” or “adjusting” the data), one wants to relate to applied work using this data to study several outcomes that you mention in the intro (i.e., development, environmental outcomes, etc.). Aside some papers about vaccination and health outcomes, there are hardly any reference to this type of papers. Think that many of your potential readers will be authors in journal like the JouEcoGeo JouUrbEco, JourDevStud, etc. I recommended trying to relate to recent work in these journals. I see no reference.

Minor:

Try to shorten (or break) sentences were possible.

Reviewer #2: The authors adequately addressed the comments of reviewers. I believe that this manuscript is now acceptable for publication.

7. PLOS authors have the option to publish the peer review history of their article (what does this mean?). If published, this will include your full peer review and any attached files.

Reviewer #1: No

Reviewer #2: No

---

## [Author Response · Author response to Decision Letter 1]

16 Mar 2022

31 January 2022

Dear Dr. Vadrevu,

Thank you for this opportunity to provide minor revisions to our manuscript, “How accurate are WorldPop-Global-Unconstrained gridded population data at the cell-level?: A simulation analysis in urban Namibia”. We have responded to comments below in italics, and made corresponding revisions to the manuscript in track changes.

Reviewer #1

1. Authors have not made all data underlying the findings in their manuscript fully available. 

It is unclear why the reviewer believes that the underlying findings are not fully available with our manuscript. All of the datasets that we used to simulate populations are publicly available and linked in the cited publication by Thomson et al. 2018 and in Table 3. The simulated outdated censuses are based on actual historical satellite imagery, which is publicly available and cited. Our parameters to define inaccurate censuses are based on a systematic literature search, which is described and cited. Finally, our simulated “true” population and all 32 versions of our simulated censuses are provided in Supplement 2. If we have missed any datasets, please let us know which ones and we will make them available or provide the corresponding links.

2. I still find the paper very technical, but I understand that this is the contribution. My concern is that, given its focus and style, the reach of the paper will be limited (see my next comment). In this line, the new intro helps. The discussion had not changed much and could try to be broader in scope. 

Related to the above, the literature continuous to be deficient. Most references are technical. For applied “users” of gridded data (rather than researchers “creating” or “adjusting” the data), one wants to relate to applied work using this data to study several outcomes that you mention in the intro (i.e., development, environmental outcomes, etc.). Aside some papers about vaccination and health outcomes, there are hardly any reference to this type of papers. Think that many of your potential readers will be authors in journal like the JouEcoGeo JouUrbEco, JourDevStud, etc. I recommended trying to relate to recent work in these journals. I see no reference.

We appreciate the push to keep data users in mind because we, ultimately, hope this paper can impact how gridded population modellers measure and report accuracy, and thus improve the accuracy and usability of gridded datasets for users. 

However, the stated focus of this paper is on a creative approach to measure fine-scale accuracy of a gridded population dataset (see last paragraph of introduction). Broadly, the implications are the same for all indicators and sectors, so we have added the following sentence to the first paragraph of the discussion, “In practical terms, this means that urban development indicators calculated with a WorldPop-Global-Unconstrained dataset at fine scale (e.g., neighbourhood) would likely be incorrect, and could lead to confusing results. For example, an underestimate of the number of people living in a neighbourhood could both make vaccination coverage rates as well as disease infection rates appear incorrectly high in that neighbourhood.” 

However, we do not intend to broaden the scope of the discussion further to specific use cases of one or more gridded population datasets because the results do not support this.

Note that in our last revision, we added discussion of how our findings about the WorldPop-Global-Unconstrained model might translate to other gridded population models if assessed for accuracy in the same way. We also cited additional urban development journals, including Environ Urban, in the opening paragraph while listing potential uses cases for fine-scale gridded population estimates.

We hope these explanations and edits are acceptable to the reviewer.

3. Try to shorten (or break) sentences were possible.

We split longer sentences in several places throughout the paper (e.g. lines 94, 104, 147, 183).

Reviewer #2: “The authors adequately addressed the comments of reviewers. I believe that this manuscript is now acceptable for publication.”

We thank both reviewers for their time and constructive feedback which has helped to strengthen the paper. Please do not hesitate to contact us with any questions or concerns.

Most sincerely,

Dana R. Thomson (with Douglas R. Leasure, Tomas Bird, Nikos Tzavidis, and Andrew J. Tatem)

---

## [Editor Report · Decision Letter 2]

3 May 2022

PONE-D-21-16343R2How accurate are WorldPop-Global-Unconstrained gridded population data at the cell-level?: A simulation analysis in urban NamibiaPLOS ONE

Dear Dr. Thomson,

Thank you for submitting your manuscript to PLOS ONE. After careful consideration, we feel that it has merit but does not fully meet PLOS ONE’s publication criteria as it currently stands. Therefore, we invite you to submit a revised version of the manuscript that addresses the points raised during the review process.

Please revise manuscript to reflect application potential of the topic with relevant references.

We look forward to receiving your revised manuscript.

Kind regards,

Krishna Prasad Vadrevu, Ph.D

Academic Editor

PLOS ONE
---

## [Author Response · Author response to Decision Letter 2]

3 Jul 2022

2 July 2022

Dear Dr. Vadrevu,

Thank you for this opportunity to provide minor revisions to our manuscript, “How accurate are WorldPop-Global-Unconstrained gridded population data at the cell-level?: A simulation analysis in urban Namibia”. Please find our responses below in italics and track changes in the manuscript.

Editor:

1. Please revise manuscript to reflect application potential of the topic with relevant references.

We have added the following paragraph to the discussion:

This analysis reinforces findings of other studies which find that currently available gridded population products tend to underestimate populations in urban areas [94–96], especially in higher-density poorer neighbourhoods [97]. For example, Tuholske and colleagues (2021) compared five gridded population products to estimate the proportion of population affected by natural disasters (SDG 11.5) in three regions where disasters had occurred, and found that 1x1 km population estimates varied widely among data products, and reflected anywhere from 20% to 80% of the total UN estimated population in each region. Furthermore, they found that WorldPop-Global-Unconstrained generally performed better than un-modelled products (e.g., GPW), but not as well as products that constrained estimates to settled cells (e.g., GHS-POP) [94]. In a separate comparison of nine gridded population estimates in Kenyan and Nigerian slum populations (SDG 11.1) where field counts were available for reference, the estimated population in each slum varied widely and WorldPop-Global-Unconstrained estimates reflected just 11% of the overall slum population while the best performing data product (HRSL) estimated just 34% of all slum dwellers [97]. A key take-away from gridded population comparison studies is that fine-scale accuracy across data products varies substantially depending on location, potentially leading to different conclusions and decisions (e.g., about the humanitarian need or health care burden) depending on the gridded population dataset used for analysis. Furthermore, these studies underscore the need to understand fine-scale accuracy across gridded population datasets and locations to inform improvements to the underlying modelling methods and inputs. 

2. Please review your reference list to ensure that it is complete and correct.

We checked the references to ensure they are complete and correct, including URL links. 

Review #1:

3. I still find the paper very technical, but I understand that this is the contribution. My concern is that, given its focus and style, the reach of the paper will be limited (see my next comment). In this line, the new intro helps. The discussion had not changed much and could try to be broader in scope. 

Related to the above, the literature continuous to be deficient. Most references are technical. For applied “users” of gridded data (rather than researchers “creating” or “adjusting” the data), one wants to relate to applied work using this data to study several outcomes that you mention in the intro (i.e., development, environmental outcomes, etc.). Aside some papers about vaccination and health outcomes, there are hardly any reference to this type of papers. Think that many of your potential readers will be authors in journal like the JouEcoGeo JouUrbEco, JourDevStud, etc. I recommended trying to relate to recent work in these journals. I see no reference.

We did not find any studies that applied or evaluated gridded population datasets in the Journal of Economic Geography, Journal of Urban Ecology, and Journal of Development Studies. We still stand by our previous response to this comment – that we do not intent to discuss “specific use cases of one or more gridded population datasets because the results do not support this.” However, as detailed above, we did add a paragraph to the discussion about other gridded population accuracy assessments and comparison studies in the contexts of disaster response (SDG 11.5) and estimating slum populations (SDG 11.1). 

4. Try to shorten (or break) sentences were possible.

We split a few additional sentences throughout the paper to improve readability (e.g., lines 52, 392) and made a few additional minor edits to improve readability.

Please do not hesitate to contact us with any questions or concerns.

Most sincerely,

Dana R. Thomson (with Douglas R. Leasure, Tomas Bird, Nikos Tzavidis, and Andrew J. Tatem)

---

## [Editor Report · Decision Letter 3]

5 Jul 2022

How accurate are WorldPop-Global-Unconstrained gridded population data at the cell-level?: A simulation analysis in urban Namibia

PONE-D-21-16343R3

Dear Dr. Thomson,

We’re pleased to inform you that your manuscript has been judged scientifically suitable for publication and will be formally accepted for publication once it meets all outstanding technical requirements.

Kind regards,

Krishna Prasad Vadrevu, Ph.D

Academic Editor

PLOS ONE
---

## [Editor Report · Acceptance letter]

13 Jul 2022

PONE-D-21-16343R3 

How accurate are WorldPop-Global-Unconstrained gridded population data at the cell-level?: A simulation analysis in urban Namibia 

Dear Dr. Thomson:

I'm pleased to inform you that your manuscript has been deemed suitable for publication in PLOS ONE. Congratulations! Your manuscript is now with our production department. 

Kind regards, 

on behalf of

Dr Krishna Prasad Vadrevu 

Academic Editor

PLOS ONE